# Rab18 Drift in Lipid Droplet and Endoplasmic Reticulum Interactions of Adipocytes under Obesogenic Conditions

**DOI:** 10.3390/ijms242417177

**Published:** 2023-12-06

**Authors:** Jaime López-Alcalá, M. Carmen Soler-Vázquez, Carmen Tercero-Alcázar, Julia Sánchez-Ceinos, Rocío Guzmán-Ruiz, María M. Malagón, Ana Gordon

**Affiliations:** 1Department of Cell Biology, Physiology, and Immunology, Adipobiology Group, Instituto Maimónides de Investigación Biomédica de Córdoba (IMIBIC), Reina Sofia University Hospital, University of Córdoba, 14004 Córdoba, Spain; b12loalj@uco.es (J.L.-A.); mcsolervazquez@ub.edu (M.C.S.-V.); b22tealc@uco.es (C.T.-A.); bc2gurur@uco.es (R.G.-R.); 2Department of Biochemistry and Physiology, School of Pharmacy and Food Sciences, Instituto de Biomedicina de la Universitat de Barcelona (IBUB), Universitat de Barcelona, 08028 Barcelona, Spain; 3Cardiology Unit, Department of Medicine-Solna, Karolinska Institute (KI), Karolinska University Hospital (NKS), 17177 Stockholm, Sweden; julia.sanchez.ceinos@ki.se; 4CIBER Fisiopatología de la Obesidad y Nutrición (CIBERobn), Instituto de Salud Carlos III, 28029 Madrid, Spain

**Keywords:** adipocyte, endoplasmic reticulum, lipid droplet, Rab18, fibrosis, inflammation

## Abstract

The adipose tissue stores excess energy in the form of neutral lipids within adipocyte lipid droplets (LDs). The correct function of LDs requires the interaction with other organelles, such as the endoplasmic reticulum (ER) as well as with LD coat-associated proteins, including Rab18, a mediator of intracellular lipid trafficking and ER–LD interaction. Although perturbations of the inter-organelle contact sites have been linked to several diseases, such as cancer, no information regarding ER–LD contact sites in dysfunctional adipocytes from the obese adipose tissue has been published to date. Herein, the ER–LD connection and Rab18 distribution at ER–LD contact sites are examined in adipocytes challenged with fibrosis and inflammatory conditions, which represent known hallmarks of the adipose tissue in obesity. Our results show that adipocytes differentiated in fibrotic conditions caused ER fragmentation, the expansion of ER–LD contact sites, and modified Rab18 dynamics. Likewise, adipocytes exposed to inflammatory conditions favored ER–LD contact, Rab18 accumulation in the ER, and Rab18 redistribution to large LDs. Finally, our studies in human adipocytes supported the suggestion that Rab18 transitions to the LD coat from the ER. Taken together, our results suggest that obesity-related pathogenic processes alter the maintenance of ER–LD interactions and interfere with Rab18 trafficking through these contact sites.

## 1. Introduction

Adipose tissue plays an essential role in energy homeostasis by managing lipid reserves in the body [1]. Adipocytes have the largest cell volume of cells in the adipose tissue and represent its defining cell type [2]. They are specialized in storing energy in the form of neutral lipids, mainly triglycerides (TGs) (i.e., lipogenesis), within their most characteristic organelles, the lipid droplets (LDs) [3]. The LD surface is coated with a variety of proteins that participate in the regulation of LD stability and other processes related to the dynamic control of lipid regulation [4], including enzymes involved in lipogenesis or lipolysis (i.e., when TGs are hydrolyzed to fatty acids and glycerol), structural proteins (e.g., the PLIN family), and the CIDE (cell death-inducing DFFA-like effector) protein family [5,6,7,8]. In addition, LDs also contain proteins related to intracellular membrane trafficking that may also mediate the interaction of LDs with other intracellular organelles, such as those of the Rab (*Ras-associated binding protein*) family of small GTPases [5,6,8,9,10]. In fact, both homo- and heterotypic interactions of LDs, such as those established with the endoplasmic reticulum (ER), mitochondria, peroxisomes, lysosomes, endosomes, and the nucleus [4,6,8,11], enable lipid traffic while preventing their free circulation inside the cells, which would cause lipotoxicity and oxidative stress [12]. In this line, perturbations of the profile of contact sites between LDs and other cellular organelles have been linked to various diseases [13,14].

Rab proteins have been classically involved in the regulation of intracellular vesicle transport, i.e., from vesicle budding in the donor compartment to their subsequent transport, anchoring, and fusion at the target membrane of the respective acceptor compartment [15,16]. Likewise, a role for Rab proteins in membrane tethering events, in combination with their interacting proteins (i.e., Rab effectors), has been also proposed [17]. Interestingly, the presence of Rab proteins in isolated LDs from different cell types has been extensively described by several proteomic studies [18,19,20]. Furthermore, the Rab family of small GTPases is one of the most abundant protein groups in the LD envelope [9,21]. According to current knowledge, Rab proteins, as other proteins of the LD coat, could target the LD in multiple ways: through cytosol–LD translocation, through diffusion from the ER to the LD, or even by interacting with proteins already bound to LDs [8,22,23]. However, few studies have been focused on deciphering how these proteins reach the LD in adipocytes [24,25,26], and none of them have delved into whether this binding is affected by the pathogenic processes accompanying obesity in this cell type.

To date, Rab18 stands out as the best-characterized LD-associated Rab protein in adipocytes and in other LD-containing non-adipocyte cell lines [27,28,29,30,31]. More specifically, we and other authors have shown that Rab18 binds to adipocyte LDs in response to either lipogenic or lipolytic stimuli induced by insulin and β-adrenergic receptor stimulation, respectively [28,29,30,31]. Previous results from our group have also shown that Rab18 mRNA and/or protein content is upregulated in subcutaneous and omental adipose tissues from obese individuals as compared to lean subjects [31]. Consistent results were observed in three different murine models of obesity (diet-induced, ob/ob, and New Zealand obese mice), where the expression levels of Rab18 increased in both visceral and subcutaneous adipose tissues [30]. However, it is still unknown as to whether the binding of Rab18 to LDs is disturbed in adipocytes under obesity conditions. Using an in vitro model of hyperglycemia/hyperinsulinemia, which induces obesity-associated insulin resistance in 3T3-L1 adipocytes, we recently showed that these conditions significantly reduced Rab18 binding to adipocyte LDs, thus indicating that high concentrations of glucose and insulin, such as those occurring in obesity, prevent Rab18 trafficking to LDs [9]. These observations were accompanied by a reduction in adipocyte functions regulated by Rab18, namely lipogenesis and/or lipolysis, in a similar manner to that shown upon Rab18 down-regulation by siRNA treatment [9,31].

It is well-known that, besides altering normal glucose and insulin circulating levels, obesity induces adipose tissue inflammation, via the secretion of proinflammatory adipokines, and fibrosis, characterized by excessive extracellular matrix deposition, leading to local and systemic metabolic dysfunctions such as insulin resistance [32]. However, no studies so far have dissected the impact of these obesity insults on Rab18’s interaction with adipocyte LDs.

Likewise, it has not been studied yet whether inflammation and fibrosis may disrupt heterotypic LD contact sites in adipocytes. In this sense, the study of the cellular interactome of LD adipocytes could be useful to identify valuable targets for obesity treatment.

Therefore, the aim of this work was to analyze the morphology and interaction of two fundamental organelles for adipocyte lipid metabolism, LDs and the ER, in response to two major hallmarks of obesity, i.e., fibrosis and inflammation.

## 2. Results

### 2.1. Fibrosis Alters the Biogenesis of Lipid Droplets and Fragments the Endoplasmic Reticulum in 3T3-L1 Adipocytes

First, we carried out a quantitative analysis of the morphometric features of LDs and evaluated the integrity of the ER in confocal images from 3T3-L1 adipocytes differentiated in three-dimensional (3D) collagen type I (COL-I) hydrogel matrices containing or not containing lumican (10 and 30 ng/mL) to mimic the fibrotic milieu found in the obese adipose tissue (Figure 1). These studies confirmed our previous observations from this in vitro model of fibrosis [9], showing that the number of LDs, immunostained with the anti-PLIN1 antibody (green signal), did not change, while the size of the LDs was significantly smaller in adipocytes that were differentiated in COL-I hydrogels containing lumican than in control matrices (i.e., COL-I alone) (Figure 1A–C).

Next, to analyze the ER structure in 3T3-L1 adipocytes grown in the 3D matrices, this organelle was specifically labeled with an ER-Tracker (red signal). Morphometric quantification of confocal images of 3T3-L1 adipocytes showed that the number of ER structures was higher in cells differentiated in lumican-containing COL-I matrices than in matrices containing COL-I alone, with this effect being dose-dependent (Figure 1A,D). In contrast, lumican dose-dependently decreased the average size of the ER structures (Figure 1E). These results indicate that the fibrotic conditions imposed by the presence of lumican caused ER fragmentation without promoting changes in the total ER mass measured as the calnexin protein content (Appendix A) and the total area of ER per cell (Appendix A). In this line, quantitative immunoblotting studies reveal a significant increase in the protein content of the ER stress marker, BIP [33], in cells exposed to 10–30 ng/mL of lumican as compared to control cells (Figure 1F).

Given the changes observed in LD size and ER fragmentation, the relationship between both LDs and ER cisternae was also evaluated to characterize ER–LD contact sites. Specifically, we evaluated the number of ER–LD contact sites, the contact area between ER fragments and LDs, and the degree of colocalization between LD and ER fluorescent markers. These studies showed that while the number of ER–LD contact sites did not change (Figure 1G), the contact surface and colocalization between LD and ER fluorescent signals increased significantly, respectively, in 3T3-L1 adipocytes grown in COL-I matrices containing 10 or 30 ng/mL of lumican as compared to control conditions (Figure 1H,I). These results, together with the morphometric data from LDs (Figure 1B,C), suggest that fibrosis may impact LD growth from the ER.

### 2.2. Fibrosis Altered ER and LD Dynamics and Hindered Rab18’s Access to Lipid Droplets

It is widely accepted that at least some of the proteins decorating the LD surface can reach the LDs via lateral diffusion from the ER, both in the small nascent LDs budding from the ER and in the larger LDs that remain connected to the ER through bridges [22,34]. To be more specific, it has been shown that, as for other Rab proteins containing a CAAX motif, Rab18 requires initial ER membrane association prior to specific organelle targeting [35]. Moreover, it has been described that Rab18 is present at the ER–LD contact sites, wherein this protein intervenes in both the increase in size of the newly formed LDs and the transfer of cargoes to these organelles [36]. These data, together with our present observations, led us to analyze the ability of Rab18 to access the LD from the ER in our experimental setting mimicking fibrosis. To this end, we performed triple-staining studies to label LDs (anti-PLIN1 antibody; green), the ER (ER-Tracker; red), and Rab18 (anti-Rab18 antibody; magenta) in 3T3-L1 cells differentiated in COL-I hydrogels (Figure 2A). The number, area, and degree of colocalization between Rab18 and the other two markers were then quantified in confocal micrographs from 3T3-L1 adipocytes grown in 3D matrices (Figure 2B–G). Regarding Rab18 binding to LDs, the number of Rab18–PLIN1 colocalizations was lower (Figure 2B), while their average area was not altered (Figure 2C) in COL-I matrices containing increasing concentrations of lumican as compared to control matrices. These latter results could be related to the lower LD area observed in between cells grown in the presence of lumican (Figure 1E). In fact, the degree of colocalization between Rab18 and PLIN1 immunofluorescence signals decreased with lumican (Figure 2D).

Regarding Rab18–ER interaction, our quantitative studies showed that the number of colocalizations tended to decrease, though not significantly, in the presence of lumican (Figure 2E). However, the area of Rab18–ER overlapping was greatly increased by lumican, especially at the higher dose tested (Figure 2F). These data are consistent with the higher degree of Rab18–ER colocalization that was observed in response to lumican (Figure 2G). When these observations are viewed together with our data showing increased ER fragmentation in adipocytes grown in the presence of this proteoglycan (Figure 1D,E), it appears that Rab18 tends to concentrate at the ER–LD contact sites.

Data on the degree of localization of Rab18 at LDs (Figure 2D), together with those for Rab18 at the ER (Figure 2G), suggest that, although Rab18 can access the LD surface, this translocation may be partially prevented, probably leaving the GTPase retained near the ER–LD contact sites. Indeed, analysis of the frequency distribution of LDs according to their size showed that Rab18 binding to LDs was lower in cells exposed to lumican than in control cells in almost all of the area intervals studied (Figure 2H). Notably, small LDs, presumably nascent and attached to the ER, were more abundant in cells exposed to high lumican concentrations (Figure 2H).

To evaluate whether the alterations observed in Rab18 distribution could be due to changes in the protein content of the GTPase, we carried out immunoblotting analysis of protein extracts that were obtained from the cells in 3D cultures. The quantification of Rab18 immunoreactive bands showed no significant differences in the intracellular content of this GTPase between control and lumican-treated adipocytes (Figure 2I).

### 2.3. Inflammatory Conditions Increased ER–LD Association

As for the 3D cell culture models, we carried out morphometric studies to characterize LDs and ER as well as ER–LD contact sites, and Rab18 distribution in 2D cultures of 3T3-L1 adipocytes exposed to tumor necrosis factor-alpha (TNFα), as a model of inflammation [37]. To this end, cells were double stained for LDs (PLIN1, green) and ER (ER-Tracker, red) (Figure 3A).

As shown in Figure 3, the exposure of 3T3-L1 adipocytes to TNFα did not change the number or the average size of LDs as compared to what was observed in the control cultures (Figure 3B,C). Likewise, the cytokine did not trigger ER fragmentation, as neither the quantity nor the area of this organelle was significantly affected by TNFα treatment (Figure 3D,E). Furthermore, TNFα exposure did not modify the content of BIP (Figure 3F) and calnexin or the total ER mass (Appendix A). Finally, no changes were observed in either the number of contact sites or the average area of ER–LD contact sites in cells treated with TNFα vs. untreated cells (Figure 3G,H), while the degree of ER–LD colocalization increased under the conditions of inflammation (Figure 3I).

### 2.4. TNFα Treatment Increased the Rab18 Localization to the ER

The interaction of Rab18 (magenta) with LDs (green) or ER cisternae (red) was investigated using the anti-Rab18 antibody and specific antibodies and/or markers for these compartments (Figure 4A). Quantification of confocal microscopy images showed that TNFα treatment did not cause significant changes in either the number and area of Rab18 localization to LDs or the degree of overlapping between Rab18 and PLIN1 immunosignals (Figure 4B–D). The number of Rab18 colocalizations with the ER and the Rab18–ER colocalization degree were also unaffected (Figure 4E,G), but the area of Rab18–ER colocalization sites was significantly increased by exposure to TNFα (Figure 4F).

As for the 3D cell model of fibrosis, we further investigated whether Rab18 trafficking to the LDs of 3T3-L1 adipocytes was affected in response to inflammatory conditions by analyzing the frequency distribution of LDs and the corresponding Rab18–LD colocalization coefficients (Figure 4H). It was observed that TNFα treatment acutely decreased the number of the smallest LDs (<3 µm^2^) and increased the number of medium and large LDs (especially 6–15 µm^2^ and 18–21 µm^2^). These findings seem to be responsible for the fact that the average size of LDs does not change (Figure 3C).

In addition, the Rab18–LD colocalization degree of the smallest LDs (3 µm^2^) in TNFα-treated cells was very low compared to basal cells. However, the Rab18–LD colocalization increased in some groups of medium and large LDs (3–6 µm^2^; 12–18 µm^2^) (Figure 4H). This trade-off regarding LD size which depends, at least in part, on Rab18–LD colocalization explains the unchanged results of the average area of Rab18–LD colocalization and Rab18–LD colocalization degree measured throughout the entire cell (Figure 4C,D). Thus, it appears that TNFα did not alter Rab18 translocation to LDs, although the ER recruited more Rab18 than under basal conditions (Figure 4F).

Quantitative immunoblotting studies showed that TNFα induced no significant changes in Rab18 intracellular content (Figure 4I).

### 2.5. Rab18 Reaches the LD Coat Mainly through the ER in Human Adipocytes

At this point, our results suggest that ER–LD contact sites are important for the translocation of proteins to the LD, including the GTPase Rab18, which also mediates such contact and, presumably, drifts like other proteins through the bridges established between LDs and the ER subdomains: (i) ER exit sites (ERES) and (ii) ERGIC [22,38]. Then, on the LD, it can perform its modulating function of lipid-metabolism dynamics [36,39].

Since previous results have shown that some treatments alter the distribution and intracellular organization of Rab18, we set out to investigate how Rab18 trafficking from the ER to the LD occurs in human cells. For this, immunocytochemistry studies were performed on human omental adipocytes from normoglycemic patients. Cells were differentiated in vitro to early (D5) and late (D10) stages and labeled with anti-Rab18 and markers of ER (ER-Tracker) or ERGIC (anti-ERGIC53), and their colocalization degrees were examined (Figure 5).

The results show that Rab18 primary colocalizes to the ER rather than to the ERGIC in human adipocytes, regardless of the differentiation state, suggesting that the Rab18’s transport from ER to LD could occur via ER–LD bridges, irrespective of whether they are LD budding points or ERES subdomains.

## 3. Discussion

In obesity, adipocytes are exposed to critical external pathological stimuli, namely fibrosis and inflammation, which, together with adipocyte hypertrophy due to lipid accumulation, induce organelle dysfunction, including mitochondria and ER damage [40,41,42]. In this line, it has been clearly established that the aforementioned conditions convey the activation of stress processes, such as oxidative stress and/or ER stress, within the adipocyte [40,41,42]. However, it is unknown how these processes affect the structure of cellular compartments and, more notably, the interactions established between them, such as those of the ER, essential for lipid metabolism, and the main organelle of the adipocyte and key compartment for lipid storage and mobilization, the LD.

Accordingly, in this work, we aimed to investigate how obesity-associated fibrosis and inflammation, which are related to the loss of lipid homeostasis in the adipose tissue [40,42], translate into intracellular changes and their influence on LD and ER interactome. In addition, the small GTPase Rab18 was employed as a molecular tool due to its association with both intracellular compartments and its location near to ER-LD contact sites [24,31,36]. Specifically, it has been reported that Rab18 at ER–LD contact sites helps to maintain homeostasis and growth of the forming LDs, keeping the latter to bind to the ER [24,36]. Furthermore, Rab18 can maintain the ER structure, at least in COS7 cells, where the redistribution of Rab18 from the ER to the cytosol triggers ER fragmentation [43].

To analyze the integrity of ER, LDs, and their interaction under fibrotic conditions, a 3D in vitro model for 3T3-L1 adipocyte differentiation was employed. In this model, we have previously shown that the proteoglycan lumican, which acts as a fibrosis inducer in adipocytes, does no alter cell viability or cytotoxicity, although it impairs adipogenesis and triggers ER stress in these cells [9]. Thus, lumican increased the protein content of BIP [9], an ER chaperone that protects cells against abnormal protein folding under stress conditions [44], without altering the total ER mass. ER stress has been linked to the onset of the fibrotic state in type-II alveolar epithelial cells and hepatocytes [45,46], and ER fragmentation in hepatocytes [44], indicating a likely vicious cycle. Interestingly, the bile acid tauroursodeoxycholic acid (TUDCA) has been shown to restore ER homeostasis in ER-stressed cells [47]. Additionally, pretreatment of human primary preadipocytes with TUDCA prevented ER stress induced by exposure to conditions mimicking the hyperglycemia/hyperinsulinemia milieu found in obesity (i.e., high concentrations of glucose and insulin), indicating a potentially useful treatment for preadipocyte damage prevention in obesity [48].

In our fibrosis cellular model, firstly, we observed ER fragmentation and smaller LD size in adipocytes in 3D cultures that were exposed to lumican. Since LD biogenesis occurs at the ER [49], this observation suggested that fibrosis, and especially the presence of lumican, could cause a change in ER–LD contact dynamics. This hypothesis was later confirmed by means of localization studies for ER and LDs performed in adipocytes grown in 3D hydrogels. These studies showed an increase in the ER–LD contact surface in response to lumican, without affecting the number of contact sites. Together, these observations indicate that the lumican-induced reduction in LD size could be due, at least in part, to the impairment of LD budding and growth triggered by fibrotic conditions favoring ER fragmentation. In view of these data, it is tempting to speculate that ER fragmentation, by reducing the ER surface area and membrane tension, could prevent newly formed LDs from budding from the ER [50].

In relation to the lipid action of the small GTPase Rab18, it has been shown that it can be temporally located at the contact points between the ER and the LDs, as described above, where it plays an important role in LD biogenesis [51]. Specifically, Rab18 has been involved in anchoring ER and nascent LDs [22] to transfer lipids synthesized by fatty acid synthase, with which Rab18 forms a complex [51]. On the other hand, Rab18 also coordinates the formation of connections between the ER and mature LDs previously separated from the ER (expanding LDs) [22] and participates in the transfer of cargos to the LD, mainly lipids and enzymes for lipid synthesis [51]. Furthermore, it has been proposed that ER–LD contact sites may be the access point of Rab18 to the forming LDs [52]. Once Rab18 reaches the LD surface, it regulates lipid storage and mobilization from the LDs in response to insulin and β-adrenergic stimulation, respectively [31]. When analyzing the colocalization of Rab18 to the ER in our fibrotic model, an increase in the overlap between Rab18 and ER markers was found in the presence of lumican. However, the number and the degree of the Rab18–LD colocalizations were significantly lower at higher lumican concentrations. These results suggest that, under the conditions of fibrosis, the translocation of Rab18 to the LD surface is partially prevented, probably due to its retention near the LD budding sites at the ER membrane, from which the small nascent LDs cannot be cleaved and subsequently grown, as in normal conditions [36]. In fact, LDs in the smallest range were more abundant in the matrices with the highest concentration of lumican. Moreover, the number of LDs decorated with Rab18 was lower in cells exposed to a high concentration of lumican, even though the Rab18 protein content remained unchanged. Taken together, our results from the fibrosis model are compatible with the reported role of Rab18 in LD growth [51]. Therefore, fibrosis-induced reduction in Rab18 at LDs may contribute, at least in part, to the inability to form larger LDs, especially in a context of ER fragmentation.

In the TNFα-induced inflammation model, a significant increase in the ER–LD colocalization degree was demonstrated with respect to the basal condition, as occurred in the fibrotic in vitro model. However, the ER–LD colocalization seemed to expand in a way that did not depend on the integrity of the ER or the LDs, as TNFα caused no effects on the number or size of ER fragments or LDs. In fact, the ER stress sensor BIP did not change its protein expression levels in response to TNFα, contrary to what occurred in the fibrotic model. Moreover, in cells treated with the cytokine, an increase in the size of the Rab18–ER colocalization zones was observed, without affecting the association of Rab18 with LDs. A deeper analysis of LD morphometric data showed that the proportion of the smallest LDs decreased in response to the inflammatory insult, while the number of medium and large LDs generally increased. In addition to these changes, smaller LDs from TNFα-treated cells had less Rab18 bound than those of the same size range in the control cells. However, while the total Rab18 protein content remained unchanged, its localization at the LDs was higher in some larger size intervals (i.e., medium and large LDs). The latter results would explain why the average number and size of Rab18–LD colocalizations were compensated and were not significantly affected. In fact, Rab18 has been also found in other cellular organelles such as the Golgi apparatus, endosomes, and peroxisomes [53,54]. Altogether, the results obtained from adipocytes challenged with pro-inflammatory conditions indicate that the avidity of Rab18 for LDs is size-dependent. These results are compatible with the best-known function of Rab18 in LDs (i.e., increasing the size of the LDs) [31,51], especially in mature LDs, where Rab18 facilitates their contact with the ER [22] to promote its growth [51]. On the other hand, Rab18 was mostly recruited to the ER, regardless of the physical communication between ER and LD. Given that under the inflammatory conditions tested, the ER structure was not affected, we could suggest that Rab18 may play a role in preserving its structure [43,54].

Finally, the translocation of Rab18 to the LD was studied throughout the differentiation of normoglycemic human adipocytes from omental adipose tissue. Thus, through immunocytochemistry, we were able to demonstrate that the preferential trafficking route of Rab18 to the LD surface is mediated by ER subdomains in an ERGIC-independent manner, as previously reported by others [51], both in early (D5) and late (D10) stages of differentiation. Supporting the latter, it has been postulated that Rab18 could access the LD through ER–LD bridges related to LD biogenesis or using the late cargo transport route to the LD, i.e., ERES [8,22].

In summary, in this work, we describe, for the first time, the impact of obesity-associated pathogenic conditions fibrosis and inflammation on the ER–LD interaction in adipocytes. In addition, our results provide further support for the role of ER–LD bridges as a possible access route for Rab18 to the LD surface in adipocytes, which may be altered in obesity conditions, as occurs upon fibrosis-induced ER fragmentation. Our studies in human adipocytes indicate that the GTPase may be transferred from the ER to the LDs. Finally, inflammatory conditions expanded ER–LD contact sites transiently—that is, without collapsing LD budding, but causing a shift in the avidity of Rab18 for LDs, depending on their size (see Graphical Abstract).

Although the stability of the ER and LDs, as well as their temporal connections, has been shown to be highly dynamic, their close physical apposition in maintaining cellular homeostasis may be critical in adipocyte functionality. Furthermore, understanding the dynamics of these contact sites may be particularly relevant to increasing our knowledge of clinical disorders related to adipose tissue pathologies that currently present poorly understood subcellular phenotypes. Indeed, most studies have been focused on finding biomarkers that change their expression levels, while the cellular localization of these biomarkers is also important, as has been demonstrated in this article.

## 4. Materials and Methods

### 4.1. 3T3-L1 Cell Cultures and Experimental Treatments

3T3-L1 cells (American Type Culture Collection, ATCC, Manassas, VA, USA) were cultured and differentiated into adipocytes according to our standard protocols for two-dimensional (2D) [31,48,55,56] or 3D cell cultures [9].

#### 4.1.1. Two-Dimensional Cell Culture of 3T3-L1 Cells and Inflammation Model

For experiments using 2D cell cultures, 3T3-L1 cells were seeded onto 3.5 cm-diameter plates or square glass coverslips (thickness #1 ½, i.e., 0.17 mm) at a density of 1800 cells/cm^2^ and differentiated in Dulbecco’s modified Eagle’s medium (DMEM) supplemented with 10% *v*/*v* fetal bovine serum (FBS), 111.5 mg/L 3-isobutyl-1-methylxanthine (IBMX), 98.1 mg/L dexamethasone, and 10 μg/mL insulin for 3 days (until day 3, D3). Thereafter, the medium was replaced by DMEM with 10% *v*/*v* FBS and 10 mg/mL insulin, and the cells were incubated for an additional 3-day period (until D6), when the medium was replaced by DMEM with 10% *v*/*v* FBS. One day later (D7), a subset of differentiated cells was incubated with pretreatment medium, consisting of DMEM with 1 g/L glucose supplemented with 1% *v*/*v* antibiotic-antimycotic solution, L-glutamine (584.6 mg/L), and 0.5% *w*/*v* bovine serum albumin (BSA) for 2 h, to synchronize the cell cycle [57,58], at 37 °C with 5% CO_2_. Thereafter, the pretreatment medium was removed and a treatment medium with the pro-inflammatory cytokine TNFα (Sigma-Aldrich, St. Louis, MO, USA) was added to generate an in vitro model of inflammation, as described previously by us [37]. This treatment consisted of DMEM with 1 g/L glucose supplemented with 1% *v*/*v* antibiotic-antimycotic solution, L-glutamine (584.6 mg/L), 0.5% *w*/*v* BSA, and mouse TNFα (86.5 µg/L) for 24 h at 37 °C with 5% CO_2_ [37]. Finally, cells at D8 were collected and processed for immunoblotting and/or confocal microscopy to analyze the effect of inflammation on adipocyte function.

#### 4.1.2. Three-Dimensional Cell Culture of 3T3-L1 Cells and Fibrosis Model

For experiments using 3D cell cultures, 3T3-L1 cells were cultured and differentiated in 3D-based COL-I (Cultrex^®^, Trevigen, Gaithersburg, MD, USA) microgels to generate an in vitro model of fibrosis using our established protocols [9]. Briefly, 3T3-L1 cells (10^5^ cells/mL) were mixed with COL-I, (3.6 mg/mL) previously neutralized with DMEM containing HEPES (0.24 mg/mL). Cells in the hydrogels were seeded onto 24-well plates or round glass coverslips (thickness #1, i.e., 0.15 mm). The mixture was allowed to gel (2 h, 37 °C) and differentiation was carried out as indicated for 2D cultures of 3T3-L1 cells, in the absence (control, COL-I) or presence of lumican (R&D System, Minneapolis, MN, USA) at 10 or 30 ng/mL (COL-I + 10 L or COL-I + 30 L), as described [9]. Thereafter, cells were collected and processed for immunoblotting, and/or confocal microscopy to analyze the effect of fibrosis on adipocyte function.

### 4.2. Human Adipocytes Cell Culture

Human primary adipocytes (preadipocytes differentiated in vitro) were obtained from omental adipose tissue samples of individuals with obesity (body mass index (BMI) > 40 kg/m^2^) undergoing bariatric surgery (see Institutional Review Board Statement and Informed Consent Statement sections) following our standard protocols [48]. Seven patients underwent a clinical assessment including their medical history, physical examination, and body composition analysis (Appendix A). Biochemical assays were carried out as previously described [37,48]. Briefly, following adipose tissue digestion using 2 mg/mL collagenase (Sigma-Aldrich, St. Louis, MO, USA), the resulting mixture was filtered through cell strainer filters with a 100 µm pore size to eliminate undispersed tissue. After washing, the preadipocytes were obtained and differentiated in vitro as previously described [48]. Briefly, cells of the stromal-vascular fraction were seeded in a 2D cell culture with preadipocyte-proliferation medium DMEM/F-12 (1:1) supplemented with 2 mg/mL biotin, 3.9 mg/mL d-pantothenate acid, 17.6 mg/mL ascorbate, 1% *v*/*v* penicillin-streptomycin, and 10% *v*/*v* new-born calf serum at 37 °C in a humidified atmosphere with 95% air/5% CO_2_. The medium was replaced every 48 h until confluency. Thereafter, cells were detached with a trypsin–EDTA solution and cultured at 4000 cells/cm^2^ three times to purify and amplify the cell culture following our established methods [48]. Preadipocytes were seeded onto glass coverslips (thickness #1 ½, i.e., 0.17 mm) at a density of 4000 cells/cm^2^ and induced for adipogenic differentiation. Cells were fixed with 4% *w*/*v* paraformaldehyde (PFA) (15 min) at D5 and D10 of differentiation. Immunostaining of these preadipocytes was carried out using our standard protocols (see Immunocytochemistry and Confocal Microscopy section).

### 4.3. Immunocytochemistry and Confocal Microscopy

3T3-L1 adipocytes or human omental adipocytes were fixed in 4% *w*/*v* PFA for 15 min at room temperature (RT), incubated with phosphate-buffered saline (PBS) containing 0.3% *w*/*v* saponin and 1% *w*/*v* bovine serum albumin (1 h at RT), and then exposed to the corresponding antibodies: (i) guinea pig anti-PLIN1 at 1:1000 dilution (Progen, Heidelberg, Germany); (ii) rabbit anti-Rab18 at 1:500 dilution (Sigma-Aldrich, St. Louis, MO, USA); and/or (iii) mouse anti-ERGIC53 at 1:100 dilution (Santa Cruz Biotechnology, Heilderberg, Germany). Thereafter, anti-guinea pig Alexa Fluor™ 488-conjugated secondary antibody alone or in combination with anti-rabbit Alexa Fluor™ 647-conjugated secondary antibody, anti-rabbit Alexa Fluor™ 488-conjugated secondary antibody, and/or anti-mouse Alexa Fluor™ 594-conjugated secondary antibody (Invitrogen, Waltham, MA, USA) were employed. In some experiments, cells were previously counterstained with ER-Tracker (for ER quantification) (Invitrogen, Waltham, MA, USA). Samples were mounted on slides with Dako Fluorescence Mounting Medium (Agilent Technologies, Santa Clara, CA, USA). For experiments using 2D cell cultures, samples were examined under a ZEISS LSM 880 super-resolution confocal laser scanning microscope equipped with an Airyscan detection unit (Carl Zeiss AG, Oberkochen, Germany).

For experiments using 3D cell cultures, samples were examined under a ZEISS LSM 710 confocal laser scanning microscope (Carl Zeiss AG., Oberkochen, Germany), since the only glass coverslips that support the processing of samples embedded in the matrix of the fibrosis in vitro model have the appropriate thickness to obtain the maximum image resolution using this microscope, and not using the microscope used in the inflammation in vitro model. After that, confocal images were processed using the Huygens Essential software package 2.4.4 (SVI, Hilversum, The Netherlands).

#### 4.3.1. Morphometric Analysis of Endoplasmic Reticulum and Lipid Droplets

Morphometric studies of the LDs and ER were performed as described with Fiji/ImageJ (NIH, USA) [48]. Different analyses were carried out by selecting from all z-stacks the middle stack for each cell separately to avoid artificial overlapping. Next, the 8-bit image was converted into a binary image that consisted of the pixels comprising the LD/ER. Following binarization, the image was subjected to watershed object separation for image processing, which was used to identify borders of adjacent LD/ER structures. After separation, the binary image was manually compared with the original image for consistency and correct binary conversion. After setting the scale of the image, (i) the holes left by the LDs were manually surrounded with the elliptical selection tool and measured in μm^2^, and (ii) the amount and individual size of the ER structures displayed in the image were measured in μm^2^. Incomplete LD/ER structures located at the edge of the image were excluded. The number and average area of all the LD/ER structures present in a cell were used to evaluate the LD/ER integrity. Images obtained from 3D and 2D cell cultures were treated equally, since each experimental group was compared with its specific control group.

#### 4.3.2. Colocalization Studies and Analysis of ER–LD Contact Sites

Colocalization of the fluorescence signals for LDs and ER cisternae was estimated by determining the overlapping pixel map of the channels (i.e., mask) using the Colocalization Finder plugin for Fiji/ImageJ 1.50b (NIH, Bethesda, MD, USA), as previously described [48,56]. The Manders’ coefficient (colocalization degree) was also assessed using the Colocalization Threshold plugin for Fiji/ImageJ (NIH, USA). Negative controls without primary or secondary antibodies were included to assess nonspecific staining. For the analysis of the number and size of contact sites/colocalization between signals, the DiAna plugin of the Fiji/ImageJ 1.50b (NIH, Bethesda, MD, USA) software was used [59]. The terminology “number and area of contact sites” was used to refer to the number and size of signals overlapping from two organelles (i.e., ER–LD), respectively. For overlapping involving the Rab18 protein (i.e., Rab18–ER and Rab18–LD), the terminology “number and area of colocalizations” was used.

### 4.4. Immunoblotting

Protein extracts were obtained from cells lysed in RIPA buffer containing 6.1 mg/mL Tris-HCl (pH 7.4), 8.8 mg/mL NaCl, 1% *v*/*v* Triton-X-100, 372.2 mg/L EDTA, and 1 μg/mL protease and phosphatase inhibitor cocktail (Waltham, MA, USA). Extracts were resuspended in loading buffer 5X (60.6 mg/mL Tris-HCl, 7.5% *w*/*v* SDS, 3.7 mg/mL EDTA, 50% *w*/*v* sucrose, 5% *v*/*v* β-mercaptoethanol, 38.6 mg/mL DTT, and 5 mg/mL bromophenol blue, pH 6.8) and heated at 97 °C for 5 min. Samples (20–30 μg) were separated by SDS-PAGE under denaturing conditions and transferred to nitrocellulose membranes (Bio-Rad Laboratories, Inc., Hercules, CA, USA) as previously described [48,55]. BIP (mouse anti-BIP at 1:1000 from Thermo Scientific, Waltham, MA, USA), calnexin (rabbit anti-calnexin at 1:1000 from Cell Signaling, Danvers, MA, USA) and Rab18 (rabbit anti-Rab18 at 1:500 from Sigma-Aldrich, Madrid, Spain) primary antibodies were dispensed overnight (4 °C), and peroxidase-conjugated secondary antibody was incubated for 1 h at RT. The immunoreaction was visualized using Clarity Western ECL Substrate (Bio-Rad Laboratories, Inc., Hercules, CA, USA). Ponceau S was selected as a loading control [48,56,60,61]. Densitometric analysis of the immunoreactive bands was carried out with Fiji/ImageJ 1.50b (NIH, Bethesda, MD, USA) software. The immunoblots shown in the manuscript included at least 3 biological replicates corresponding to the complete set of samples (control and experimental groups) that were run on the same gel. The original uncropped blots of specific figures are provided as Appendix A.

### 4.5. Statistical Analysis

Statistical analysis was carried out using GraphPad Prism 8 statistical software, (GraphPad Software). Prior to each analysis, outliers were identified and eliminated using the ROUT method (Q = 1%). The normal distribution of variables was assessed using the Shapiro–Wilk test. One-way ANOVA, Kruskal–Wallis tests, independent *t*-tests, or Mann–Whitney tests were used where appropriate. A post hoc statistical analysis using Tukey’s or Dunn’s test was performed to identify significant differences between groups. Values were considered significant at *p* < 0.05. For confocal imaging assays, at least 10 micrographs from two biological replicates were analyzed. The number of micrographs per condition and all raw image analyses are shown in Appendix A. For the immunoblotting assays, at least three biological replicates were analyzed.

## 5. Conclusions

Obesity-associated fibrotic conditions may cause ER fragmentation in adipocytes.Both fibrosis and inflammation increased the number of ER–LD contact sites, but only the former hindered LD biogenesis and growth.The fibrotic environment may affect Rab18 trafficking to the LD.TNFα promotes Rab18 accumulation at the ER in adipocytes, which may contribute to the maintenance of ER structure.Studies of Rab18 in human (pre)adipocytes support the route followed by this GTPase from the ER to LDs in 3T3-L1 adipocytes.

## Figures and Tables

**Figure 1 ijms-24-17177-f001:**
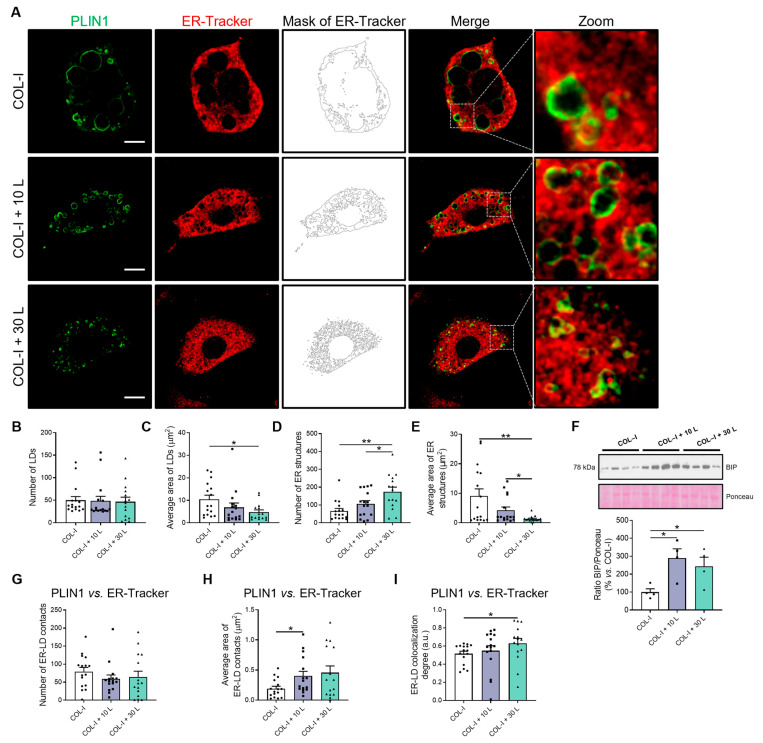
Morphometry of LDs, ER, and ER–LD contact sites in adipocytes under fibrosis conditions. (**A**) Representative confocal microscopy images of 3T3-L1 cells at D8 of differentiation, cultured on COL-I gels in the absence of lumican (Control/COL-I, 0 ng/mL lumican) or in the presence of 10 or 30 ng/mL lumican (COL-I + 10 L and COL-I + 30 L, respectively). Cells were stained with anti-PLIN1 (green) and ER-Tracker (red). (**B**,**D**) Number and (**C**,**E**) average area of LDs or ER structures per cell. (**F**) Representative immunoblot of BIP protein content in 3T3-L1 cell extracts. Ponceau staining was used as loading control (n = 4 per condition). (**G**) Number, (**H**) average area, and (**I**) colocalization degree of ER–LD overlapping. (**B**–**E**,**G**–**I**) Graphs show the mean ± SEM (n ≥ 15 cells per condition from 2 independent experiments). Scale bar, 10 μm. (**F**) One-way ANOVA and Tukey’s tests. (**B**–**E**,**G**–**I**) Kruskal–Wallis and Dunn’s tests. * *p* < 0.05; ** *p* < 0.01.

**Figure 2 ijms-24-17177-f002:**
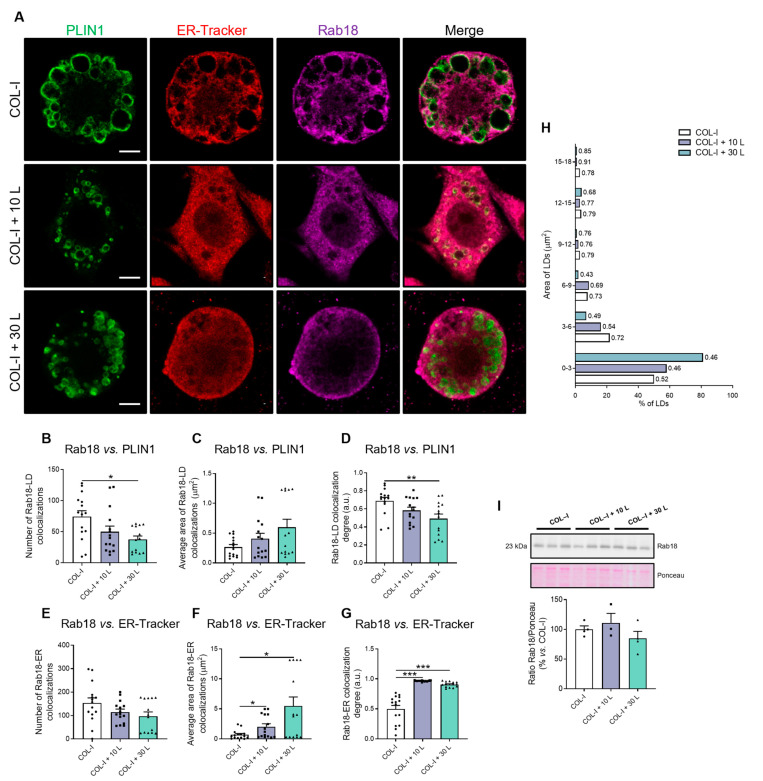
Rab18 overlapping with ER-LD interaction in adipocytes under fibrosis conditions. (**A**) Representative confocal microscopy images of 3T3-L1 cells at D8 of differentiation, cultured on COL-I gels in the absence of lumican (Control/COL-I, 0 ng/mL lumican) or in the presence of 10 or 30 ng/mL lumican (COL-I + 10 L and COL-I + 30 L, respectively). Cells were stained with anti-PLIN1 (green), ER-Tracker (red) and anti-Rab18 (magenta). (**B**,**E**) Number, (**C**,**F**) average area and (**D**,**G**) colocalization degree of Rab18–LD or Rab18–ER overlapping. (**H**) Morphometric analysis of LDs’ frequency distribution. Manders’ coefficient of Rab18–LD overlapping is shown to the right of each bar. (**I**) Representative immunoblot of Rab18 protein content in 3T3-L1 cell extracts. Ponceau staining was used as loading control (n = 3–4 per condition). (**B**–**G**) Graphs show the mean ± SEM (n ≥ 14 cells per condition from 2 independent experiments). Scale bar, 10 μm. (**G**) One-way ANOVA and Tukey’s tests. (**B**–**F**) Kruskal–Wallis and Dunn’s tests. * *p* < 0.05; ** *p* < 0.01; *** *p* < 0.001.

**Figure 3 ijms-24-17177-f003:**
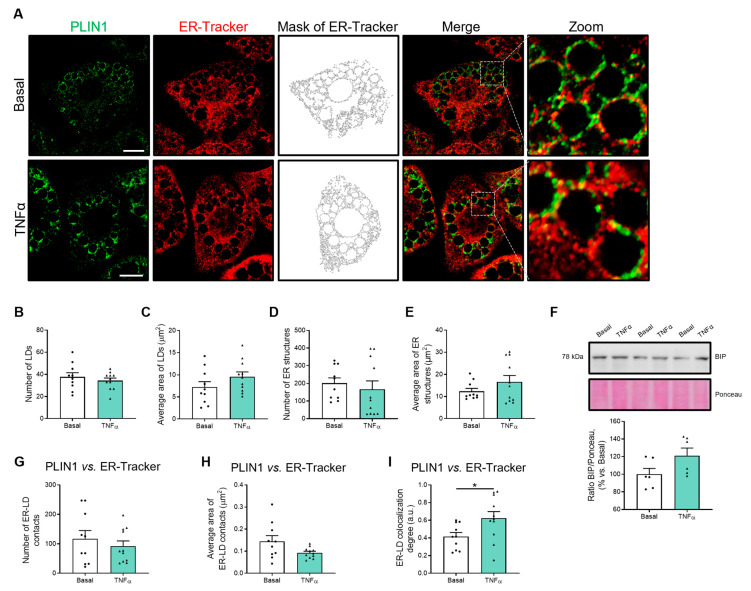
Morphometry of LDs, ER, and ER–LD contact sites in adipocytes under inflammatory conditions. (**A**) Representative confocal microscopy images of 3T3-L1 cells at D8 of differentiation, under basal conditions or treated with TNFα (86.5 µg/L; 24 h). Cells were stained with anti-PLIN1 (green) and ER-Tracker (red). (**B**,**D**) Number and (**C**,**E**) average area of LDs or ER structures per cell. (**F**) Representative immunoblot of BIP protein content in 3T3-L1 cell extracts. Ponceau staining was used as loading control (n = 6). (**G**) Number, (**H**) average area, and (**I**) colocalization degree of ER–LD overlapping. (**B**–**I**) Graphs show the mean ± SEM (n ≥ 10 cells per condition from 2 independent experiments). Scale bar, 10 μm. (**B**,**C**,**F**–**I**) Independent *t*-test. (**D**,**E**) Mann–Whitney test. * *p* < 0.05.

**Figure 4 ijms-24-17177-f004:**
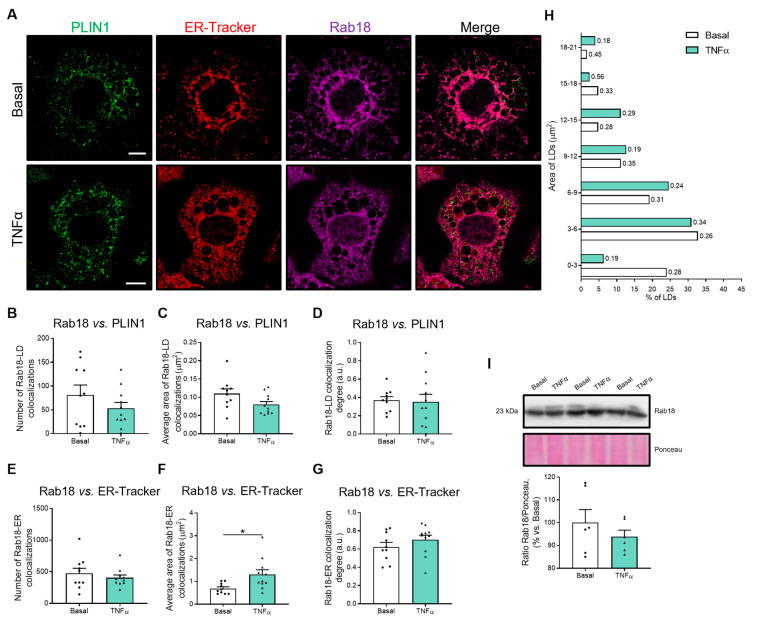
Rab18 overlapping with ER-LD interaction in adipocytes under inflammation conditions. (**A**) Representative confocal microscopy images of 3T3-L1 cells at D8 of differentiation, under basal conditions or treated with TNFα (86.5 µg/L; 24 h). Cells are stained with anti-PLIN1 (green), ER-Tracker (red) and anti-Rab18 (magenta). (**B**,**E**) Number, (**C**,**F**) average area and (**D**,**G**) colocalization of Rab18-LD or Rab18-ER overlapping. (**H**) Morphometric analysis of LDs frequency distribution. Manders’ coefficient of Rab18-LD overlapping is shown to the right of each bar. (**I**) Representative immunoblot of Rab18 protein content in 3T3-L1 cell extracts. Ponceau staining was used as loading control (n = 6). (**B**–**G**) Graphs show the mean ± SEM (n ≥ 10 cells per condition from 2 independent experiments). Scale bar, 10 μm. (**B**–**G,I**) Independent *t*-test. * *p* < 0.05.

**Figure 5 ijms-24-17177-f005:**
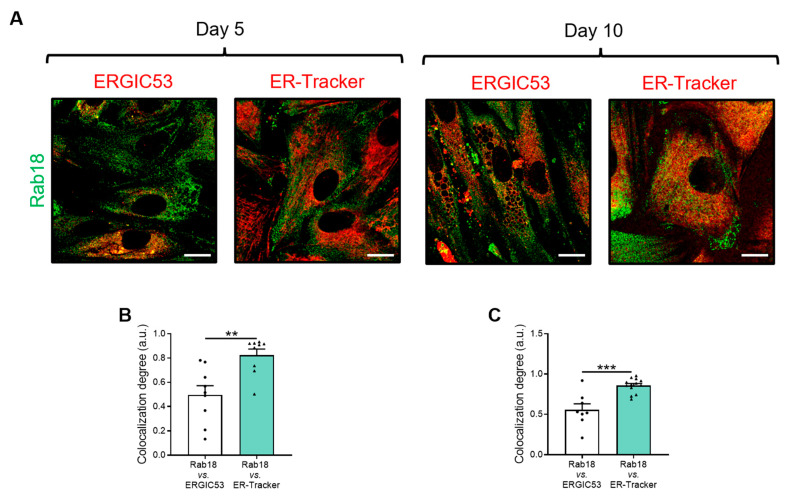
Analysis of access routes of Rab18 to LDs. (**A**) Representative confocal microscopy images of human omental adipose tissue-derived adipocytes from normoglycemic patients, at D5 and D10 of differentiation. Cells are stained with anti-ERGIC53 or ER-Tracker (red) and with anti-Rab18 (green). (**B**) Colocalization of Rab18 with ERGIC53 or ER-Tracker at D5 and (**C**) D10 of differentiation. Graphs show the mean ± SEM (n = 9 cells per condition). Scale bar, 10 μm. (**C**) Independent *t*-test. (**B**) Mann–Whitney test. **, *p* < 0.01; ***, *p* < 0.001.

## Data Availability

All the raw data and their corresponding analyzed images presented in this study are available in Appendix A and Appendix A.

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
