# Peer review of "Rab18 Drift in Lipid Droplet and Endoplasmic Reticulum Interactions of Adipocytes under Obesogenic Conditions"

_ijms, 2023, doi:10.3390/ijms242417177_

Round 1
Reviewer 1 Report
Comments and Suggestions for Authors
Jaime López-Alcalá et al. conducted a fascinating study exploring the relationship between Rab18, lipid droplets, and endoplasmic reticulum interactions in adipocytes under obesogenic conditions. While providing valuable data, the manuscript has several concerns that need to be addressed before being considered for publishing.
-
Need to evaluate the ER stress levels under the fibrotic and inflammatory conditions.
-
Based on the figures and analytical results, the measurements of the morphology of endoplasmic reticulum and lipid droplets are very objective. Please add more data points and provide the raw data in supplements.
-
Evaluate the ER volume using WB.
-
Describe the statistical method used in each figure in the legend.
Comments on the Quality of English Language
The manuscript is clearly illustrated in English.
Author Response
Comments and Suggestions for Authors
Jaime López-Alcalá et al. conducted a fascinating study exploring the relationship between Rab18, lipid droplets, and endoplasmic reticulum interactions in adipocytes under obesogenic conditions. While providing valuable data, the manuscript has several concerns that need to be addressed before being considered for publishing.
We do appreciate the thorough review and constructive suggestions offered by the Reviewer to improve the quality and impact of our paper. Accordingly, we have edited the English language and made the revisions detailed below.
1. Need to evaluate the ER stress levels under the fibrotic and inflammatory conditions.
The reviewer is correct that endoplasmic reticulum (ER) stress levels were not completely specified under both studied conditions. While the microscopic ER fragmentation was analyzed for both pathological cell models, the measurement of the protein levels of the ER stress marker, BIP (Binding-Immunoglobulin Protein), was only determined for 3T3-L1 adipocytes exposed to TNFα. To justify the latter, in the previous version of the manuscript, we stated that the BIP chaperone content was previously reported by our laboratory in Guzmán-Ruiz et al., FASEB J, 2020 (doi:10.1096/fj.201902703R). However, we understood that to address the Reviewer’s comment and also to validate the results obtained by microscopic analysis of ER stress associated with fibrosis, we had to perform a BIP immunoblot specific to the current manuscript. Thus, the quantification of BIP bands confirmed similar conclusions to those obtained in Guzmán-Ruiz et al., FASEB J., 2020 (doi:10.1096/fj.201902703R): there was a significant increase in the BIP content of COL-I + 10 L and COL-I + 30 L cells compared to control cells (COL-I). Both the Western blot membrane and Ponceau visualization as well as the uncropped scan membranes have been added to Figure 1F and new Figure S2, respectively. These new results have been included in lines 125-128. Finally, to relate the ER fragmentation to the ER stress marker BIP in a TNFα model, the immunoblot has been moved from Figure 4I to the new Figure 3F and the results included in lines 234-235 of the new version of the manuscript.
2. Based on the figures and analytical results, the measurements of the morphology of endoplasmic reticulum and lipid droplets are very objective. Please add more data points and provide the raw data in supplements.
We do thank the Reviewer for supporting our analytical results. Following the Reviewer’s suggestion, we have made our analyses more precise and clearer to the reader by changing the bar graphs to scatter plots and changing the figure legends to include the exact number of cells measured in each experimental condition. This involved an increase in the number of data points of some of the analyzed groups, as proposed by the reviewer. Additionally, the raw data from these imaging analyses were added to new Table S2 (referred to lines 576-577 of the new version of the manuscript).
3. Evaluate the ER volume using WB.
We really appreciate this interesting question, since the evaluation of the total ER volume or mass can provide more information about the functionality of this organelle, especially when it is fragmented (higher number of ER structures and less size of them), as shown in Figures 1A, D, and E. The experienced reader may wonder about this phenomenon. Notably, changes in the ER volume can involve from the collapse of the Golgi/ERGIC transport pathways to the ER (Smirle et al., Cold Spring Harb Perspect Biol, 2013, doi: 10.1101/cshperspect.a015073; (Rainer-Duden, Mol Membr Biol, 2003, doi: 10.1080/0968768031000122548) to the action of autophagy mechanisms of damaged ER (ER-phagy) (Yang, et al., 2021, Front Cell Dev Biol, doi: 10.3389/fcell.2021.684526).
To delve into this question, the Reviewer has recommended inferring the ER volume by Western blot. Following these indications, we have used an antibody against calnexin, the integral membrane protein of this organelle, as the ER marker (Stone et al., J Virol, 2007, doi: 10.1128/JVI.01366-06). These new results show the preservation of total ER mass (determined by the quantification of calnexin immunoblots from cellular protein extracts), under both obesity-derived fibrotic and inflammatory conditions, compared to their corresponding control groups (results included in new Figure S1 and new Figure S2). Furthermore, the results from measuring the total size of the ER as the sum of the areas of all the ER structures within each cell also support the above findings (results included in new Figure S1 and new Figure S2). That is, as shown by the immunoblot results, the total ER content did not change significantly in any of the experimental groups of the two pathophysiological conditions analyzed. Although the ER integrity analysis showed changes in the number and average size of the ER structures (i.e., ER fragmentation) in the obesity-derived fibrosis model (Figures 1A, D, E), our new results also suggest that the ER fragmentation could occur without an acute alteration in the turnover or recycling of intracellular membranes, at least at the time of evaluation. These new results have been included and discussed in lines 123-125, 234-235 and 327-329 of the new version of the manuscript.
4. Describe the statistical method used in each figure in the legend.
The reviewer is right that in the previous version of the manuscript we did not describe the statistical method figure by figure in the corresponding legends. We apologize for this oversight. We have done our best to clarify this issue in all figure legends in the new version of the manuscript.
Reviewer 2 Report
Comments and Suggestions for Authors
In this manuscript, the authors have investigated the fibrosis and inflammation conditions affecting the budding of LD from ER, including the role of Rab18 in between. This is a very interesting project that concentration on the size of LD, level of ER-LD interaction within obesity related pathogenic conditions.
Here are the comments:
In result 2.1 and Figure 1, I suggest the authors plot the raw data points in the bar figure, which can help the readers to have a more obvious view of the lumican effect on tested parameters. Also, I think it could be a good option for the authors to have a brief discussion, maybe one or two sentences, to introduce the analyzed the number of ER-LD contacts and area of contacts. Because the analysis observation can be very different between a 3D-cell analysis and 2D “flat-cell” image analysis (as you “smash” the 3D cell into 2D image, you could lose some of the contacts sites). In Figure 1-G, it looks significant between group COL-I and CLO-I + 30L, while there is no significant star labeled. I suggest the authors to have a table present all the statistic analysis data, so as the p-values.
In Figure 2-C, and Figure 4-C, what are the purposes of having a broken y axis?
I also noticed that in your cell staining images, the shape of shown cells varies. May I ask if this variation is due to the heath condition of each cell? Or if this is due to the cell being under different cell cycle stages? How did the authors exclude the influence of tested cell health condition / cell cycle on the result outcomes?
Considering the authors present good amount of imaging analysis data, The quality of each analyzed cells becomes critical. I believe all analyzed data should have their own saved images, which are stored by the authors. The authors can make the manuscript way more convincible by showing each analyzed data parallel to its representative image (maybe as supplemental figure).
Author Response
Comments and Suggestions for Authors
In this manuscript, the authors have investigated the fibrosis and inflammation conditions affecting the budding of LD from ER, including the role of Rab18 in between. This is a very interesting project that concentration on the size of LD, level of ER-LD interaction within obesity related pathogenic conditions.
We sincerely appreciate the insightful issues raised by the Reviewer. The following are responses to the specific comments, which we hope will clarify the conclusions of the results presented.
1. Here are the comments: In result 2.1 and Figure 1, I suggest the authors plot the raw data points in the bar figure, which can help the readers to have a more obvious view of the lumican effect on tested parameters.
We thank the Reviewer for this insightful comment. We agree that plotting the raw data points in bar charts would be helpful for the reader to understand the magnitude of the changes shown. We have incorporated this suggestion not only to those results related to lumican effects on the parameters tested, but also in the case of TNFα. Therefore, we have applied these changes to all graphs (Figures 1-5 and S1) and included the raw data in a new table (Table S2).
2. Also, I think it could be a good option for the authors to have a brief discussion, maybe one or two sentences, to introduce the analyzed the number of ER-LD contacts and area of contacts. Because the analysis observation can be very different between a 3D-cell analysis and 2D “flat-cell” image analysis (as you “smash” the 3D cell into 2D image, you could lose some of the contacts sites).
We thank the Reviewer for this insightful observation and fully agree with her/him that adding a short explanation to introduce the number of ER-LD contacts analyzed and the area of the contacts would help to understand the analysis performed (lines 517-518 in the new version of the manuscript). In our case, we employed the same methodology for image analysis in both the experiments on fibrosis and inflammation. To be more specific, after completion of the experiments, we performed the immunocytochemical assays and obtained the confocal microscopic images from the cells in the cultures, including all the z stacks from each cell. Thereafter, from all the z stacks, we selected the medium stack for each cell separately and analyzed within that stack the different parameters (e.g., number of ER-LD contacts and the area of the contacts); the images and corresponding raw data have been included in the new Table S2 that has been prepared for the revised version of the manuscript. The latter process was repeated for each cell. We did not fuse/superimpose all the images from the cells into a 2D image, as merging the stacks would lead to artifactual overlapping for contact studies. We employed this approach since it is commonly used in studies on colocalization studies (Guzmán-Ruiz et al., FASEB J, 2020; doi:10.1096/fj.201902703R; Travez et al., J Cell Mol Med, 2018; doi: 10.1111/jcmm.13840; Almabouada et al., J Biol Chem, 2013; doi: 10.1074/jbc.M112.404624). Furthermore, each experimental design included its own specific control group, that we used to compare the results obtained specifically with the fibrosis or the TNFα model. Nevertheless, we have indicated this limitation in the revised version of the manuscript (lines 528-530 of the new version).
3. In Figure 1-G, it looks significant between group COL-Iand CLO-I + 30L, while there is no significant star labeled. I suggest the authors to have a table present all the statistic analysis data, so as the p-values.
In Figure 1G, the standard error of the mean (SEM) of the experimental group COL-I + 30L is bigger than that of COL-I + 10L. For the latter reason, only COL-I + 10L was significantly different from the control group. Besides, the statistical test we have employed in this comparison (three groups) was Kruskal-Wallis (followed by Dunn’s test), which is more restrictive than a t-test, requiring a smaller SEM to yield statistical significance. However, following the Reviewer’s recommendation and for the sake of clarity, the new raw data table that has been generated (Table S2) includes the p-values. We thank the Reviewer for this constructive critique and apologize for the oversight.
4. In Figure 2-C, and Figure 4-C, what are the purposes of having a broken y axis?
We thank the Reviewer for his/her appreciation. Broken axes in Figure 2C (lumican-induced fibrosis) and Figure 4C (TNFα-induced inflammation) were employed for comparison purposes, that is, to highlight that the relative amount of Rab18 associated with the ER is higher than that at LDs. In the new version of the manuscript, the broken axes have been removed to accommodate the changes derived from replacing the original bar graphs by scattered plots (i.e., to avoid missing data points within the broken part of the axes).
5. I also noticed that in your cell staining images, the shape of shown cells varies. May I ask if this variation is due to the heath condition of each cell? Or if this is due to the cell being under different cell cycle stages? How did the authors exclude the influence of tested cell health condition / cell cycle on the result outcomes?
The Reviewer is correct about the images shown from COL-I + 10 L and COL-I + 30 L groups, where cells have a different shape than those of the other groups (Figure 1A). This phenomenon occurs because fibrotic conditions (lumican at high concentrations) hamper adipogenesis, that is, preventing the conversion of elongated fibroblasts to round adipocytes. These results were depicted in our previous work (Guzmán-Ruiz et al., FASEB J, 2020, doi:10.1096/fj.201902703R), showing that 30 ng/ml lumican did not change the LD number but affected the LD content. The microscopic images shown in Figures 1B and C of the current manuscript reproduce the latter results. Furthermore, previous RT-qPCR results showed that adipocytes grown in 3D cultures containing lumican expressed lower levels of the adipogenic transcription factors, C/EBPα and PPARγ, supporting the results obtained by microscopic imaging. Also, as an optimization step and to show that the effects of fibrosis were not due to an effect on cell viability, in our previous work we also performed two different assays, MTT and lactate dehydrogenase activity, and no changes in cell viability or cytotoxicity were observed under any lumican condition tested (Guzmán-Ruiz et al., doi:10.1096/fj.201902703R).
In all experiments, cells were exposed to pretreatment medium, so that the complete medium supplemented with 10% v/v fetal bovine serum (FBS) was replaced with serum-free medium, which induces the so-called "serum starvation cell cycle". In the experiments on inflammation, this was also used to prevent the influence of potential growth factors in the serum and ensure that the effects observed on differentiating cells were due to TNFa administration. In the case of adipocytes cultured in 3D matrices mimicking fibrotic conditions, which involves exposure of cells to lumican throughout differentiation, the chronic use of serum-deprived media is not compatible with cell survival.
We have included these critical considerations in lines 324-327 and 443-444 of the new version of the manuscript.
6. Considering the authors present good amount of imaging analysis data, The quality of each analyzed cells becomes critical. I believe all analyzed data should have their own saved images, which are stored by the authors. The authors can make the manuscript way more convincible by showing each analyzed data parallel to its representative image (maybe as supplemental figure).
We thank the Reviewer for her/his comment. Thus, as suggested and for the sake of clarity, in this new version we have included the raw data from all the image analyses in the new Table S2 (referred to lines 576-577 and 593 of the new version of the manuscript), as well as all the representative images for each data point (cell by cell).
Round 2
Reviewer 1 Report
Comments and Suggestions for Authors
None
Reviewer 2 Report
Comments and Suggestions for Authors
No further comments from me.